# Impact of Salvage Surgery following Colonic Endoscopic Polypectomy for Patients with Invasive Neoplasia

**Xiangzhou Tan** [1,2] , **Markus Quante** [2] , **Zihua Chen** [1] , **Zhikang Chen** [1] , **Alfred Königsrainer** [2] and **Dörte Wichmann** [2,*]

[1] Department of General Surgery, Xiangya Hospital, Central South University, Changsha 410008, China; xiangzhou.tan@csu.edu.cn (X.T.); zihuachen@csu.edu.cn (Z.C.); 403445@csu.edu.cn (Z.C.)

[2] Interdisciplinary Endoscopy Unit, Department of General, Visceral and Transplantation Surgery, University Hospital Tübingen, 72076 Tübingen, Germany; markus.quante@med.uni-tuebingen.de (M.Q.); alfred.koenigsrainer@med.uni-tuebingen.de (A.K.)

[*] Correspondence: doerte.wichmann@med.uni-tuebingen.de; Tel.: +49-70-712-968-143

**Abstract:** Background: Invasive neoplasia (Tis-T1) are increasingly being encountered in the daily routine of endoscopic polypectomy. However, the need for salvage surgery following endoscopic therapy for invasive neoplasia is controversially discussed. Patients and Methods: Patients with endoscopic removal of invasive neoplasia were identified from the national Surveillance Epidemiology and End Results (SEER) Database 2005 to 2015. Survival analysis and Cox proportional hazard regression analysis in cancer-specific mortality and overall survival rate was used, which were stratified by T stage and polyp size. Results: A total of 5805 patients with endoscopic removal of invasive neoplasia were included in the analysis, of whom 1214 (20.9%) underwent endoscopic treatment alone and 4591 (79.1%) underwent endoscopic resection plus surgery. The survival analysis revealed that patients undergoing salvage surgery had a significantly better cancer-specific survival (97.4% vs. 95.8%, *p*-value = 0.017). In patients with T1 stage, additional salvage surgery led to a significantly higher cancer-specific survival (92.1% vs. 95.0%, *p* value = 0.047). Conclusion: Salvage surgery following endoscopic polypectomy may improve the oncological survival of patients with invasive neoplasia, especially in patients with T1 stage. Furthermore, the T stage, size, and localization of polyps, as well as the level of CEA, could be identified as significant predictors for lymphonodal and distant metastases.

**Keywords:** colorectal neoplasia; endoscopic resection; salvage surgery

## 1. Introduction

Colorectal polyps are regarded as a precursor lesion for colorectal cancer (CRC) [1]. It has been widely proven that endoscopic polypectomy, as the gold standard of treatment modalities for patients with colorectal polyps, can reduce the incidence and mortality of CRC [2,3]. With further development and refinement of endoscopic techniques and devices, the polyp detection rate has been continuously increased to a current maximum of one-third of routine colonoscopies [4]. Here, 0.8% to 5.6% of colorectal polyps which were endoscopically removed in general diagnostic practice were finally confirmed as malignant polyps [5]. However, the further therapy management of patients who underwent endoscopic resection of invasive neoplasia is still controversially discussed due to the potential risk of residual tumor, lymph node metastasis, and distant metastasis among patients with invasive lesions [6].

It is widely accepted to refer CRC patients who underwent endoscopic polypectomy in T2–T4 stage for additional surgical resection. However, the controversial debate arises from invasive neoplasia with the stages of intramucosal carcinoma (Tis) and submucosal carcinoma (T1). Intramucosal carcinoma is independent of invasive CRC, which has a

negligible risk of lymphovascular metastasis [7]. The utilization of terms such as carcinoma or cancer should, thus, be avoided for lesions with Tis stage, which might otherwise lead to unnecessary surgery [8]. Submucosal carcinoma, specifically known as malignant polyps, is classified as pT1 in the TNM classification system. These lesions are able to involve lymphatic and/or hematogenous spread. The standard treatment for patients who underwent endoscopic removal of polyps with T1 stage is, therefore, either referral for surgical resection or intensified surveillance according to the risk classification. The unfavorable factors in these patients include deep submucosal invasion (>1 mm), lymphovascular invasion, poor tumor differentiation, tumor budding, and a positive resection margin [9]. However, current recommendations are based on expert consensus, which depends on indirect evidence, i.e., the risks of lymphovascular metastasis. Therefore, a population-based study that directly explores survival outcomes is urgently needed.

In this study, we aimed to compare the cancer-specific survival mortality and survival among the patients with invasive neoplasia between different treatment strategies, i.e., endoscopic polypectomy alone versus endoscopic polypectomy plus surgical resection. The clinicopathologic characteristics of patients following endoscopic removal of invasive neoplasia were also evaluated to explore potential predictive factors of lymphovascular metastasis.

## 2. Materials and Methods

### 2.1. Study Design

A long-term and large-population cohort study was conducted using the Surveillance, Epidemiology, and End Results (SEER) database to determine the efficacy of surgical resection following the endoscopic polypectomy among patients with stage 0 and stage 1 CRC. The SEER Program collects and publishes population-based data on cancer cases via 19 geographically disparate registries, which includes more than 11 million cancer cases, with an increase of around 516,000 new cases per year. The SEER database (2000–2018, updated on November 2020) was accessed using SEER Stat 8.3.9 software on 3 May 2021.

### 2.2. Study Population

All patients with invasive neoplasia referred for endoscopic polypectomy as the first stage of therapy between January 2005 and December 2015 were retrospectively evaluated for enrollment in the study, resulting in a total of 10,994 patients. Those patients who met the following exclusion criteria were removed: without positive histology or cytology confirmed; T2–4 stage MCPs; unknown TNM stage; with tumor deposits; with multiple in situ/malignant tumors; with benign/borderline tumors; where therapeutic modalities could not be inferred, i.e., the surgical status following endoscopic polypectomy. The flowchart for patient selection is shown in Figure 1.

### 2.3. Variable Definitions

Individual characteristics, including sex, age, histological grading, year of diagnosis, the localization of primary site, TNM staging, the diagnostic methods of tumor size and CS extension, the size of polyps, carcinoembryonic antigen (CEA), cancer-specific survival (CSS), and survival months were extracted from the database. Right colon refers to those invasive neoplasia proximal to the splenic flexure, while left colon includes the sigmoid colon and descending colon. The TNM staging system follows the sixth edition of the American Joint Committee on cancer staging. All patients with explicit records of surgical resection status for invasive polyps were eligible for the enrollment, and then divided into two groups: ET alone group ("no surgical resection performed") and ET plus surgery group ("surgical resection performed").

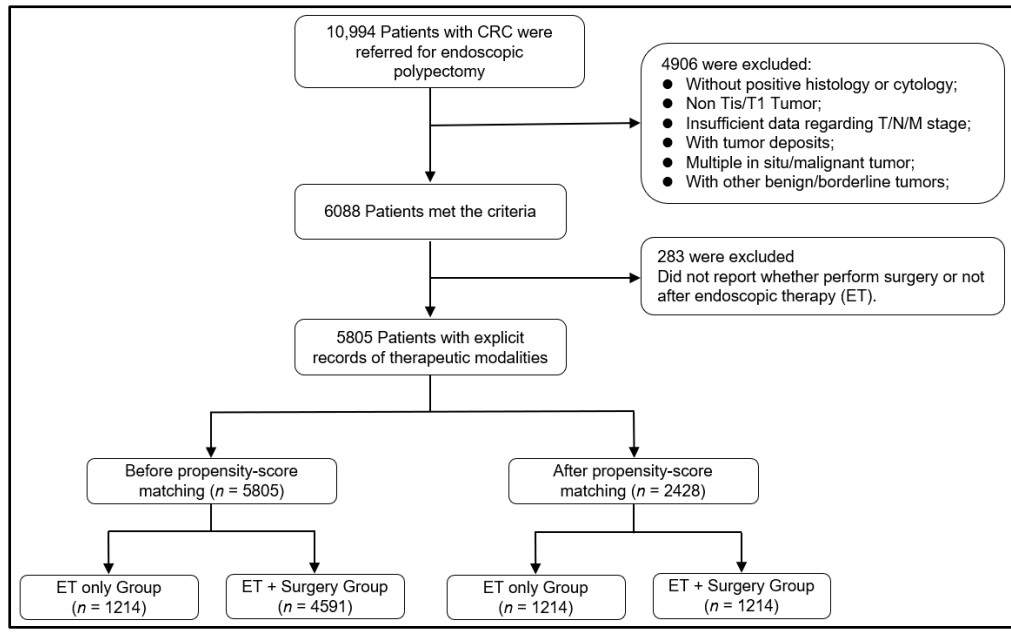

**Figure 1.** Flow chart of study enrollment and exclusions.

## 2.4. Clinical Endpoints

The primary outcome of this study was cancer-specific survival. CSS was calculated as the time from diagnosis of invasive neoplasia to death attributed to colorectal cancer or survival at last follow-up. The secondary outcome was overall survival (OS). OS was calculated as the time from diagnosis of invasive neoplasia to death attributed to any reason at last follow-up. Subgroups were stratified by T stage and the polyp size. The data for survival analysis were collected over 5 years. Furthermore, the metastasis of lymph nodes and distant tissue/organs was assessed across the entire included population. The population trend of therapy modalities by year was also exhibited.

## 2.5. Statistical Analysis

A propensity score matching (PSM) was performed between the ET alone group and ET plus surgery group with a fixed ratio of 1:1 using the nearest-neighbor matching method. The matched covariates included sex, age, histological grade, histology type, localization, T stage, and polyp size. N stage and M stage were excluded as covariates because of the potential bias caused by the TNM stage correction after the surgery. We use the chi-square and *t*-tests for categorical variables and continuous variables, respectively, to detect the distribution of variables. The CSS and OS survival were analyzed using the Kaplan–Meier method and log-rank test. A subgroup analysis according to T stage and polyp size was performed. A univariate Cox proportional hazard regression model was applied to determine the prognostic effects of T stage and polyp size on CSS survival. R software (version 3.6.0) was used to perform the data analysis. A *p*-value <0.05 was considered statistically significant.

## 2.6. Ethics

All data analyzed in this study were publicly available from the Surveillance, Epidemiology, and End Results (SEER) database (https://seer.cancer.gov (access on 3 May 2021)). No ethical approval and patient consent were required.

## 3. Results

### 3.1. Patient Characteristics

A total of 5805 patients in the SEER database were included in the cohort study. Among those, 1214 (20.9%) underwent ET alone, and 4591 (79.1%) patients underwent ET

plus surgery. A total 2428 patients were finally included for the survival analysis and Cox regression analysis after PSM. The baseline population characteristics before and after PSM are listed in Table 1. The ET alone group had a significantly older age (mean, 64.8; SD, 11.97) than the ET plus surgery group (mean, 63.23; SD, 12.06). Additionally, the distribution of polyp size between groups was different; for example, the ET plus surgery group has more cases where the polyp size was less than 10 mm (21.4% in ET alone group vs. 23.0% in ET plus surgery group).

**Table 1.** Baseline characteristics before and after propensity score matching analysis.

| Characteristics | Before PSM | | | | After PSM | | | |
|---|---|---|---|---|---|---|---|---|
| | Overall | ET Only | ET + Surg | *p*-Value | Overall | ET Only | ET + Surg | *p*-Value |
| *N* | 5805 | 1214 | 4591 | | 2428 | 1214 | 1214 | |
| Sex (%) | | | | | | | | |
| Female | 2410 (41.5) | 485 (40.0) | 1925 (41.9) | 0.226 | 989 (40.7) | 485 (40.0) | 504 (41.5) | 0.457 |
| Male | 3395 (58.5) | 729 (60.0) | 2666 (58.1) | | 1439 (59.3) | 729 (60.0) | 710 (58.5) | |
| Age (mean (SD)) | 63.41 (12.05) | 64.08 (11.97) | 63.23 (12.06) | 0.03 | 64.21 (11.87) | 64.08 (11.97) | 64.35 (11.76) | 0.573 |
| Grade (%) | | | | | | | | |
| Well-differentiated; Grade I | 793 (13.7) | 157 (12.9) | 636 (13.9) | 0.556 | 315 (13.0) | 157 (12.9) | 158 (13.0) | 0.453 |
| Moderately differentiated; Grade II | 1457 (25.1) | 290 (23.9) | 1167 (25.4) | | 566 (23.3) | 290 (23.9) | 276 (22.7) | |
| Poorly differentiated; Grade III | 121 (2.1) | 25 (2.1) | 96 (2.1) | | 51 (2.1) | 25 (2.1) | 26 (2.1) | |
| Undifferentiated, anaplastic; Grade IV | 72 (1.2) | 18 (1.5) | 54 (1.2) | | 27 (1.1) | 18 (1.5) | 9 (0.7) | |
| Unknown | 3362 (57.9) | 724 (59.6) | 2638 (57.5) | | 1469 (60.5) | 724 (59.6) | 745 (61.4) | |
| Histology type (%) | | | | | | | | |
| Adenocarcinoma | 5654 (97.4) | 1176 (96.9) | 4478 (97.5) | 0.092 | 2361 (97.2) | 1176 (96.9) | 1185 (97.6) | 0.733 |
| Mucinous adenocarcinoma | 23 (0.4) | 5 (0.4) | 18 (0.4) | | 7 (0.3) | 5 (0.4) | 2 (0.2) | |
| Neuroendocrine carcinoma | 16 (0.3) | 8 (0.7) | 8 (0.2) | | 15 (0.6) | 8 (0.7) | 7 (0.6) | |
| Signet ring cell carcinoma | 4 (0.1) | 2 (0.2) | 2 (0.0) | | 4 (0.2) | 2 (0.2) | 2 (0.2) | |
| Squamous cell carcinoma | 4 (0.1) | 1 (0.1) | 3 (0.1) | | 1 (0.0) | 1 (0.1) | 0 (0.0) | |
| Multiply histology types | 2 (0.0) | 0 (0.0) | 2 (0.0) | | 0 (0.0) | 0 (0.0) | 0 (0.0) | |
| Unknown | 102 (1.8) | 22 (1.8) | 80 (1.7) | | 40 (1.6) | 22 (1.8) | 18 (1.5) | |
| Localization (%) | | | | | | | | |
| Right colon | 1282 (22.1) | 276 (22.7) | 1006 (21.9) | 0.804 | 545 (22.4) | 276 (22.7) | 269 (22.2) | 0.678 |
| Left colon | 4407 (75.9) | 915 (75.4) | 3492 (76.1) | | 1842 (75.9) | 915 (75.4) | 927 (76.4) | |
| Rectum | 116 (2.0) | 23 (1.9) | 93 (2.0) | | 41 (1.7) | 23 (1.9) | 18 (1.5) | |
| T stage (%) | | | | | | | | |
| Tis | 3496 (60.2) | 760 (62.6) | 2736 (59.6) | 0.061 | 1497 (61.7) | 760 (62.6) | 737 (60.7) | 0.358 |
| T1 | 2309 (39.8) | 454 (37.4) | 1855 (40.4) | | 931 (38.3) | 454 (37.4) | 477 (39.3) | |
| T1, Submucosa invasion positive | 1062 (18.3) | 203 (16.7) | 859 (18.7) | | 402 (16.6) | 203 (16.7) | 199 (16.4) | |
| T1, Submucosa invasion negative | 1186 (20.4) | 227 (18.7) | 959 (20.9) | | 494 (20.3) | 227 (18.7) | 267 (22.0) | |
| Polyp size (%) | | | | | | | | |
| <10 mm | 1315 (22.7) | 260 (21.4) | 1055 (23.0) | 0.009 | 519 (21.4) | 260 (21.4) | 259 (21.3) | 0.466 |
| 10–19 mm | 404 (7.0) | 65 (5.4) | 339 (7.4) | | 121 (5.0) | 65 (5.4) | 56 (4.6) | |
| 20–29 mm | 197 (3.4) | 40 (3.3) | 157 (3.4) | | 77 (3.2) | 40 (3.3) | 37 (3.0) | |
| 30–39 mm | 81 (1.4) | 19 (1.6) | 62 (1.4) | | 27 (1.1) | 19 (1.6) | 8 (0.7) | |
| 40–49 mm | 26 (0.4) | 4 (0.3) | 22 (0.5) | | 8 (0.3) | 4 (0.3) | 4 (0.3) | |
| 50-mm | 31 (0.5) | 13 (1.1) | 18 (0.4) | | 27 (1.1) | 13 (1.1) | 14 (1.2) | |
| Unknown | 3751 (64.6) | 813 (67.0) | 2938 (64.0) | | 1649 (67.9) | 813 (67.0) | 836 (68.9) | |

## 3.2. Survival Analysis

After PSM, survival analysis according to CSS and OS was performed among the patients with invasive neoplasia (both Tis and T1 stage). Overall, the ET plus surgery group had a significantly better CSS than the ET alone group, with a higher 5 year CSS rate (97.4% in the ET plus surgery group vs. 95.8% in the ET only group, *p*-value = 0.017, Figure 2A). There was no significant difference in OS between the ET plus surgery and ET alone group (*p*-value = 0.76, Supplementary Figure S1).

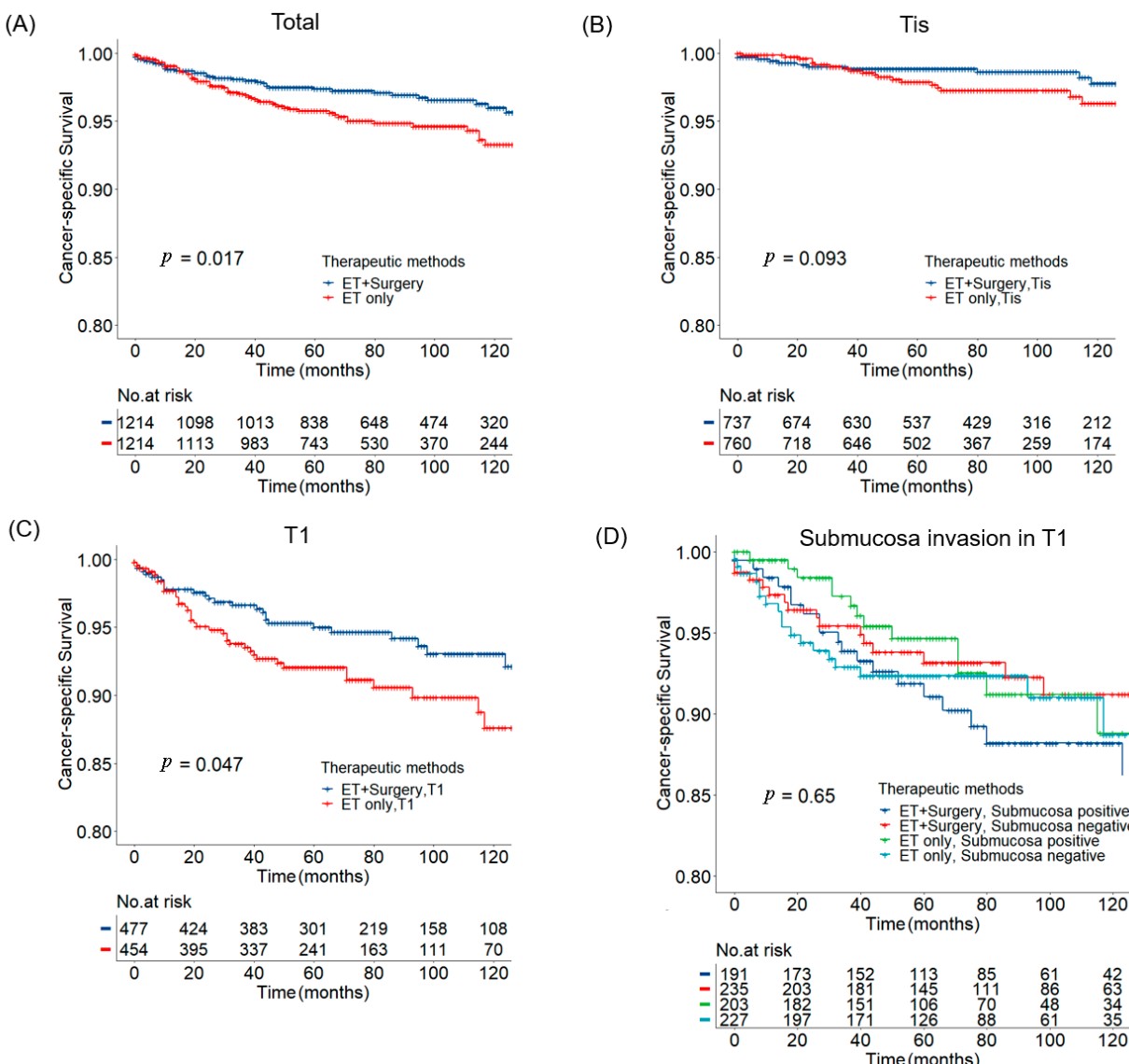

**Figure 2.** Kaplan–Meier cancer-specific survival plots in patients who underwent endoscopic removal of invasive neoplasia. (**A**) All patients; (**B**) patients with intramucosal carcinoma (Tis); (**C**) patients with submucosal carcinoma (T1); (**D**) patients with or without submucosa invasion.

Next, subgroup analysis of T stage revealed no significant difference in CSS between the ET alone group and ET plus surgery group in patients with Tis stage ($p$-value = 0.093, Figure 2B). In those patients with T1 stage, however, the 5 year CSS rate in the ET alone group was significantly lower than that in the ET plus surgery group (92.1% vs. 95.0%, $p$-value = 0.047, Figure 2C). No significant difference in CSS was observed in the subgroups of different submucosa invasion status ($p$-value = 0.65, Figure 2D). Similarly, there was no significant difference in OS in the subgroups of different T stages (Supplementary Figure S1).

In the subgroup analysis of the polyp size, there was no significant difference in CSS between the two treatment modalities among all subgroups (Figure 3A–C), as well as OS (Supplementary Figure S2). In detail, the 5 year CSS rates in the subgroup of polyp size less than 10 mm were 98.3% vs. 97.4% in the ET alone group and ET plus surgery group, respectively ($p$-value = 0.610, Table 2). The 5 year CSS rates in the subgroup of polyp size equal to 10–19 mm were 97.6% in the ET alone group vs. 96.0% in the ET plus surgery group ($p$-value = 0.770, Table 2). The 5 year CSS rates in the subgroup of polyp size more than 20 mm were 88.2% and 94.8% in the ET alone group and ET plus surgery group, respectively ($p$-value = 0.200, Table 2).

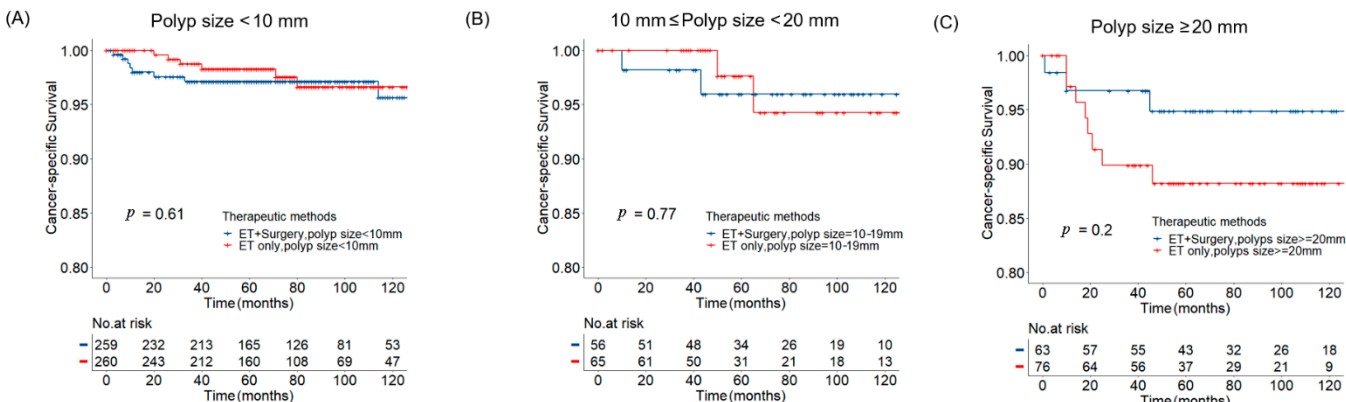

**Figure 3.** Kaplan–Meier cancer-specific survival plots in patients underwent endoscopic removal of invasive neoplasia. (**A**) Invasive polyp size less than 10 mm; (**B**) invasive polyp size between 10 mm and 20 mm; (**C**) invasive polyp size larger than 20 mm.

**Table 2.** The 5 year cancer-specific survival for different treatment modalities among patients with endoscopic removal of invasive neoplasia.

| Variable | ET Only Group | | ET + Surgery Group | | *p*-Value |
|---|---|---|---|---|---|
| | 5 Year CSS Rate | 95% CI | 5 Year CSS Rate | 95% CI | |
| Overall | 0.958 | 0.946–0.970 | 0.974 | 0.964–0.983 | 0.017 |
| T stage | | | | | |
| Tis stage | 0.979 | 0.968–0.990 | 0.988 | 0.981–0.996 | 0.093 |
| T1 stage | 0.921 | 0.894–0.948 | 0.950 | 0.929–0.971 | 0.047 |
| Submucosa invasion | | | | | |
| T1, Submucosa invasion positive | 0.947 | 0.913–0.981 | 0.950 | 0.918–0.907 | 0.552 |
| T1, Submucosa invasion negative | 0.924 | 0.888–0.960 | 0.949 | 0.920–0.978 | 0.949 |
| Polyp size | | | | | |
| <10 mm | 0.983 | 0.966–1.000 | 0.971 | 0.950–0.992 | 0.610 |
| 10–19 mm | 0.976 | 0.931–1.000 | 0.960 | 0.906–1.000 | 0.770 |
| ≥20 mm | 0.882 | 0.808–0.962 | 0.948 | 0.893–1.000 | 0.200 |

*3.3. Cox Regression Analysis*

The univariate Cox proportional hazard regression model shows that the additional surgical resection of invasive neoplasia was a significant prognostic factor, indicating that patients with surgery after endoscopic polypectomy have a better prognosis in terms of CSS (HR 0.61, 95% CI 0.40–0.92, *p*-value = 0.018, Table 2). However, the significant prognostic effect of surgical resection cannot be seen among patients with Tis stage. In contrast, additional surgery following ET could be identified as a significant prognostic factor in patients with T1 stage (HR 0.61, 95% CI 0.37–1.00, *p*-value = 0.050, Table 3). In the subgroup analysis of polyp size, ET plus surgery was not a prognostic factor among all subgroups (*p*-value = 0.607, 0.766, 0.146, in the polyp size <10 mm, 10–19 mm, and ≥20 mm, respectively).

**Table 3.** Univariate Cox regression analysis.

| Variable | | Hazard Ratio | Lower 95% CI | *p*-Value |
|---|---|---|---|---|
| Overall | ET alone | Reference | | |
| | ET + Surgery | 0.61 | 0.40–0.92 | 0.018 |
| T stage | | | | |
| Tis | ET alone | Reference | | |
| | ET + Surgery | 0.54 | 0.26–1.12 | 0.099 |
| T1 | ET alone | Reference | | |
| | ET + Surgery | 0.61 | 0.37–1.00 | 0.050 |
| Submucosa invasion | | | | |
| T1, Submucosa invasion positive | ET alone | Reference | | |
| | ET + Surgery | 0.72 | 0.32–1.64 | 0.435 |
| T1, Submucosa invasion negative | ET alone | Reference | | |
| | ET + Surgery | 0.71 | 0.36–1.40 | 0.322 |
| Polyp size | | | | |
| <10 mm | ET alone | Reference | | |
| | ET + Surgery | 1.32 | 0.46–3.81 | 0.607 |
| 10–19 mm | ET alone | Reference | | |
| | ET + Surgery | 0.76 | 0.13–4.60 | 0.766 |
| ≥20 mm | ET alone | Reference | | |
| | ET + Surgery | 0.20 | 0.02–1.76 | 0.146 |

### 3.4. Nodal Involvements and Distant Spread

To explore the probability of lymphonodal spread and distant migration among patients who underwent endoscopic polypectomy, the risk of lymph node involvement and distant migration within subcategories of T stage, polyp size, localization, and CEA level are summarized in Table 4. In total, 22/5805 (0.38%) patients who underwent endoscopic polypectomy were identified with lymphonodal involvement, while 27/5805 (0.46%) patients who underwent ET were detected with positive distant migration. Of note, no metastasis of lymph nodes and no distant tissue/organ metastasis were found among patients with Tis stage. However, a significant increase in both lymph node involvements and distant migration could be identified along with increasing polyp size. Here, nearly 10% of patients with a polyp size larger than 50 mm suffered from metastasis of lymph nodes and/or distant tissue/organs. With regard to the localization of polyps, invasive neoplasia in the rectum showed the highest risk for positive lymph nodes and distant metastasis, accounting for 1.7% and 2.6% with positive N stage and M stage, respectively.

A predictive factor for spread of disease is the occurrence of invasive polyps in the right colon, with 0.5% and 1.2% risk of positive metastasis in lymph nodes and distant tissue/organs, respectively. In contrast, invasive polyps in the left colon showed the lowest risk at only 0.3% and 0.4% risk of positive lymph nodes and distant migration. Lastly, elevated levels of CEA were associated with a higher probability of both lymph nodes and distant migration.

**Table 4.** The migration of lymph nodes and distant tissue/organs across all patients with endoscopic polypectomy.

| | Subgroup | Overall | N0 | Probability | N Stage N1 + N2 | Probability | *p*-Value | Overall | M0 | M Stage Probability | M1 | Probability | *p*-Value |
|---|---|---|---|---|---|---|---|---|---|---|---|---|---|
| *n* | | 5805 | 5783 | 99.6210% | 22 | 0.3790% | | 5805 | 5768 | 99.3626% | 37 | 0.6374% | |
| T stage | Tis | 3496 | 3496 | 100.0000% | 0 | 0.0000% | <0.001 | 3496 | 3496 | 100.0000% | 0 | 0.0000% | <0.001 |
| | T1 | 2309 | 2287 | 99.0472% | 22 | 0.9528% | | 2309 | 2272 | 98.3976% | 37 | 1.6024% | |
| T1, submucosa invasion | Submucosa positive | 402 | 400 | 99.5025% | 2 | 0.4975% | 0.289 | 402 | 400 | 99.5025% | 2 | 0.4975% | 0.041 |
| | Submucosa negative | 494 | 490 | 99.1903% | 4 | 0.8097% | | 494 | 482 | 97.5709% | 12 | 2.4291% | |
| Polyp size in mm | <10 | 1315 | 1313 | 99.8479% | 2 | 0.1521% | <0.001 | 1315 | 1314 | 99.9240% | 1 | 0.0760% | <0.001 |
| | 10–19 | 404 | 403 | 99.7525% | 1 | 0.2475% | | 404 | 400 | 99.0099% | 4 | 0.9901% | |
| | 20–29 | 197 | 195 | 98.9848% | 2 | 1.0152% | | 197 | 194 | 98.4772% | 3 | 1.5228% | |
| | 30–39 | 81 | 78 | 96.2963% | 3 | 3.7037% | | 81 | 78 | 96.2963% | 3 | 3.7037% | |
| | 40–49 | 26 | 25 | 96.1538% | 1 | 3.8462% | | 26 | 25 | 96.1538% | 1 | 3.8462% | |
| | >50 | 31 | 28 | 90.3226% | 3 | 9.6774% | | 31 | 28 | 90.3226% | 3 | 9.6774% | |
| Localization | Unknown | 3751 | 3741 | 99.7334% | 10 | 0.2666% | 0.047 | 3751 | 3729 | 99.4135% | 22 | 0.5865% | <0.001 |
| | Right colon | 1282 | 1275 | 99.4540% | 7 | 0.5460% | | 1282 | 1267 | 98.8300% | 15 | 1.1700% | |
| | Left colon | 4407 | 4394 | 99.7050% | 13 | 0.2950% | | 4407 | 4388 | 99.5689% | 19 | 0.4311% | |
| | Rectum | 116 | 114 | 98.2759% | 2 | 1.7241% | | 116 | 113 | 97.4138% | 3 | 2.5862% | |
| CEA | Negative | 424 | 420 | 99.0566% | 4 | 0.9434% | <0.001 | 424 | 421 | 99.2925% | 3 | 0.7075% | <0.001 |
| | Positive | 120 | 112 | 93.3333% | 8 | 6.6667% | | 120 | 95 | 79.1667% | 25 | 20.8333% | |
| | Unknown | 5261 | 5251 | 99.8099% | 10 | 0.1901% | | 5261 | 5252 | 99.8289% | 9 | 0.1711% | |

*3.5. Time Trend Analysis*

The time trend analysis demonstrated no significant change in the population number of patients who underwent endoscopic polypectomy ($p_{trend}$ = 0.079, Table 5). However, a significant increase in the proportion of patients who underwent ET alone was noted over time ($p_{trend}$ < 0.001). Conversely, a decreasing trend was found in the proportion of patients who underwent surgery after ET ($p_{trend}$ < 0.001).

**Table 5.** Time trend analysis of the use of salvage surgery for patients underwent endoscopic removal of invasive neoplasia.

| Year of Diagnosis | Overall | ET Alone | Incidence | ET + Surgery | Incidence |
|---|---|---|---|---|---|
| Overall | 5805 | 1214 | 20.91% | 4591 | 79.09% |
| 2005 | 485 | 80 | 16.49% | 405 | 83.51% |
| 2006 | 586 | 109 | 18.60% | 477 | 81.40% |
| 2007 | 544 | 92 | 16.91% | 452 | 83.09% |
| 2008 | 534 | 94 | 17.60% | 440 | 82.40% |
| 2009 | 521 | 100 | 19.19% | 421 | 80.81% |
| 2010 | 501 | 96 | 19.16% | 405 | 80.84% |
| 2011 | 477 | 114 | 23.90% | 363 | 76.10% |
| 2012 | 536 | 119 | 22.20% | 417 | 77.80% |
| 2013 | 554 | 132 | 23.83% | 422 | 76.17% |
| 2014 | 549 | 139 | 25.32% | 410 | 74.68% |
| 2015 | 518 | 139 | 26.83% | 379 | 73.17% |
| $p_{trend}$ | 0.079 | | <0.001 | | <0.001 |

**4. Discussion**

With the development of improved endoscopic techniques and devices, endoscopic therapy has become a widely accepted therapy for the removal of colorectal polyps, even in malignant cases. Several retrospective cohort studies have reported that endoscopic polypectomy alone is an adequate therapy for those CRC patients with low-risk factors [10,11]. Nevertheless, surgical resection remains the primary and most effective treatment for localized, advanced CRCs [12]. However, the appropriate therapy approach for colorectal cancer with stage 0 and stage 1, as well as the patient management following endoscopic removal of invasive neoplasia, remains a matter of debate.

Here, our large population-based study compared long-term CSS mortality and survival between the ET alone group and the ET plus surgery group among patients who underwent endoscopic polypectomy. As a result, our study demonstrated that subse-

quent surgical resection of corresponding colorectal segments had a significantly better CSS survival among patients with endoscopic removal of invasive neoplasia. However, there was no significant difference in CRC survival between different treatment modalities groups among those patients with Tis stage. The Cox proportional regression analysis further proved that surgical resection was a favorable prognostic factor in patients with T1 stage following endoscopic polypectomy, while such a prognostic effect could not be seen in patients with Tis stage. In addition, no subgroups according to polyp size showed significant differences in CSS between different treatment strategies in both cancer-specific survival analysis and Cox proportional regression analysis, thus indicating that polyp size alone could not independently serve as a stratification factor for risk level or as a prognostic factor.

In general, patients with intramucosal carcinoma (Tis) are supposed to have no risk of tumor recurrence or metastasis after complete endoscopic removal of these lesions. However, in our study, the 5 year cancer-specific mortalities in the ET alone group and the ET plus surgery group were 2.1% and 1.2%, respectively. This phenomenon raises the main question as to why cancer-specific deaths exist in patients with Tis stage, especially several years after the diagnosis. The primary reason could be the uncertain presence of residual tumor. A recent systematic review pointed out that the incomplete resection rate (IRR) of colorectal polyps from 1–20 mm was 13.8% when performed by snare polypectomy [13]. The IRR for forceps removal of polyps is even worse compared to snare polypectomy in polyps with 1–5 mm size (9.9% vs. 4.4%) [13]. Pedersen et al. also reported that incomplete polyp resection is common in routine clinical practice [14]. The uncertain presence of residual tumor could also explain the wide range of HRs in the Tis subgroup (Table 2). Furthermore, a small HR favored by salvage surgery can also be found among patients with Tis stage, albeit with no statistical significance (HR 0.54, *p* value = 0.099). The second explanation could be that new-onset colorectal cancer subsequent to the primary procedure may contribute to cancer-specific death. We assumed that, although the primary lesion was removed, similar unfavorable factors pre-procedure/surgery, such as genetic susceptibility, diet habit, and intestinal environment, can induce colorectal carcinogenesis, which may also count as cancer-specific death. Thirdly, the potential reason could be misclassification of tumor extension, especially in the ET alone group, because surgical resection shows advantages in the accurate staging [15].

The factors involving nodal and distant metastasis are of significant importance, which could lead to the different risk stratification and subsequently impact current treatment strategies of patients undergoing endoscopic polypectomy. In this study, the correlations between nodal/distant metastases and covariates in patients undergoing endoscopic polypectomy were analyzed. Our study illustrated that all of the included covariates, including T stage, polyp size, localization of polyps, and CEA status, were associated with both nodal involvements and distant metastasis (*p*-value < 0.05). The covariates herein can be used to estimate the risk stratification. Although the polyp size has been proven as a higher risk factor of superficial submucosal invasion [8,16], studies rarely reported the relationship between polyp size and nodal involvements and distant metastasis. The risk of both nodal involvement and distant metastasis dramatically increases with polyp size, especially when polyp size is larger than 2 cm, which is similar to the observation in early gastric adenocarcinoma [17]. Technically, those lesions larger than 20 mm in size usually require endoscopic submucosal resection or surgery, instead of en bloc endoscopic mucosal resection [9,18]. A critical decision on procedure and management is, therefore, required when treated with large invasive neoplasia. Interestingly, we found that invasive neoplasia is more often located in the left colon, which is contradictory to current expert consensus [8]. This discrepancy might arise from differences in the population (e.g., Asia vs. America) and treatment modalities (e.g., ET vs. surgery) [14]. Moreover, rectal polyps are linked to the highest risk of nodal involvement and distant metastasis, followed by right-sided polyps, with left-sided polyps at lowest risk. Of note, the significance of CEA levels in the risk assessment of patients with invasive polyps is underestimated [8,9]. Only

9.2% of patients who underwent endoscopic polypectomy completed the blood test of CEA, among which those with positive results had a considerably higher risk of both nodal and distant metastases.

The time trend analysis revealed an increasing trend of "adventure strategies", i.e., with endoscopic treatment alone in cases with invasive neoplasia. Contrarily, the total number of patients who underwent endoscopic treatment has not changed over time. Endoscopic treatment, as an innovative minimally invasive procedure, is supposed to have advantages due to its lower cost, fewer complications, shorter hospital stay, etc. However, in cases of invasive neoplasia, endoscopic treatment requires careful consideration.

In summary, possible complications after endoscopic resection are the misidentification of the T stage and polyp size, as well as periprocedural complications such as bleeding and perforation. Complications after surgical resection of Tis and T1 malignancies are the overextended operation with subsequent functional impairments, as well as possible long-term complications of the hospital stay and the operation mode such as surgical-side infections or incisional hernia. These postsurgical, short-term, and long-term complications were not the subject of the present study.

Of note, there were several limitations in our study. Firstly, our study was a retrospective study, which could have led to inevitable bias. Secondly, some of the important factors associated with the survival analysis were lacking, such as the depth of invasion and tumor budding. Lastly, the endoscopic techniques used for the removal of polyps were not disclosed in the database, such as cold/hot snare polypectomy, forceps removal, and endoscopic submucosal resection, which could have also impacted the results.

## 5. Conclusions

Salvage surgery following endoscopic polypectomy is recommended for patients with invasive neoplasia, especially for those with T1 stage. Here, the favorable prognostic effect of salvage surgery might be due to the complete en bloc resection of invasive polyps. However, endoscopic therapy without subsequent surgery for invasive neoplasia has become more and more popular. Here, radical surgical resection should be the treatment of choice if patients have unfavorable factors or if risk management has not been fully evaluated. Lastly, the T stage, size, and localization of polyps, as well as the CEA level, were identified as significant predictors for lymph node metastasis and distant metastasis.

**Supplementary Materials:** The following supporting information can be downloaded at: https://www.mdpi.com/article/10.3390/curroncol29050255/s1, Figure S1: Kaplan-Meier overall survival plots in patients underwent endoscopic removal of invasive neoplasia. (A) All patients; (B) Patients with intramucosal carcinoma (Tis); (C) Patients with submucosal carcinoma (T1). (D) Patients with or without submucosa invasion; Figure S2: Kaplan-Meier overall survival plots in patients underwent endoscopic removal of invasive neoplasia. (A) Invasive polyp size less than 10 mm; (B) Invasive polyp size between 10 mm and 20 mm; (C) Invasive polyp size larger than 20 mm.

**Author Contributions:** Conceptualization, methodology, and software, X.T., Z.C. (Zihua Chen) and D.W.; data curation and investigation: X.T., M.Q. and Z.C. (Zhikang Chen); writing—original draft preparation, X.T., M.Q. and D.W.; supervision, A.K.; writing—reviewing and editing, with critical revision of the manuscript, D.W., A.K., M.Q. and Z.C. (Zihua Chen). All authors have read and agreed to the published version of the manuscript.

**Funding:** This work was partly supported by the program of China Scholarships Council (201806370236). OA publishing is supported by the Open-Access Publication Fund of the University of Tübingen.

**Institutional Review Board Statement:** All data analyzed in this study were publicly available, from the Surveillance, Epidemiology, and End Results (SEER) database (https://seer.cancer.gov accessed on 3 May 2021). No ethical approval was required.

**Informed Consent Statement:** Patient consent was waived due to that all data analyzed in this study were publicly available, from the Surveillance, Epidemiology, and End Results (SEER) database.

**Data Availability Statement:** All data analyzed in this study were publicly available, from the Surveillance, Epidemiology, and End Results (SEER) database (https://seer.cancer.gov accessed on 3 May 2021). The detailed data presented in this study are available on request from the corresponding author.

**Conflicts of Interest:** The authors declare no conflict of interest. The funders had no role in the design of the study; in the collection, analyses, or interpretation of data; in the writing of the manuscript, or in the decision to publish the results.

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
