# Peer review of "Impact of Salvage Surgery following Colonic Endoscopic Polypectomy for Patients with Invasive Neoplasia"

_curroncol, doi:10.3390/curroncol29050255_

Round 1

Reviewer 1 Report

Thanks for giving me the possibility to review this paper. The paper address a very interesting aspect in the treatment of early phases of colo-rectal cancer. The presentation arises some dubts. 

The main is related to the staging of the patients who underwent endoscopic treatment. It is hard to understand why cN+ patients did not receive ET + surgery. It can be related to clinical factors, as well as for M1 patients. I should expect to consider only cN0, cM0 patients and consider clinical nodal and distant metastases as exclusion criteria. 

Another unclear aspect is related to tumor deposits. Tumor deposits are defined as  foci of carcinoma separated from the main lesion and identified in pericolonic or perirectal fat or the adjacent mesentery. In TNM Ed 7 and 8 tumor deposits are classified as N1c, and they can be diagnosed only with pathological examination of a surgical specimen. Probably the term is used with a different meaning. 

Author Response

Thanks for giving me the possibility to review this paper. The paper addresses a very interesting aspect in the treatment of early phases of colo-rectal cancer. The presentation arises some dubts.

-The main is related to the staging of the patients who underwent endoscopic treatment. It is hard to understand why cN+ patients did not receive ET + surgery. It can be related to clinical factors, as well as for M1 patients. I should expect to consider only cN0, cM0 patients and consider clinical nodal and distant metastases as exclusion criteria.

Response: Thanks for the constructive suggestion. We totally understand the concern of the reviewer. After our careful consideration, we still insisted that the patients with N+ or M+ should be included in the study, the reasons are listed below:

  1. The staging of lymph nodes(LNs)/Metastasis(M) herein is either clinical stage or pathological stage. That means the N stage and M stage were diagnosed or corrected after surgery. Therefore, TNM stage was potentially modified after the intervention. If we exclude those patients with N+ and M+, while some of them may be diagnosed as N0/M0 before surgery. This will artificially increase the selection bias.
  2. According to the experience of our institution, a systematic check for the status of lymph node and distant metastasis may not be a routine for the patients with invasive polyps in some hospitals. For some extents, the N/M staging might not be so accurate that to differentiate the patients with lymph node or distal metastasis. For example, those patients with N+ or M+ may fail to be diagnosed because a systematic CT scan (including full abdominal and chest CT) wasn’t applied.
  3. One of the goals in this manuscript is to explore the predictive factors of lymphovascular metastasis. If we exclude those patients with N+ and M+, the goal regarding the predictive factor of lymphovascular metastasis can’t be achieved.

PS:  According to the common sense, those patients with Tis or T1(without deep submucosa invasion >1000μm) are not supposed to develop lymph nodes or distal metastasis. But many cases with T1 stage in our study developed N+ (22/2309, 0.9528%) or M+ (37/2309, 1.6024%). To our knowledge, this may be the clinical reasons, such as the presence of residual tumor, or the quality of pathological tissue is too low (even missing) to accurately evaluate the staging, especially for the endoscopic resected tissue. We should objectively treat this phenomenon, and the problem of residual tumor in clinics did surely exist.

-Another unclear aspect is related to tumor deposits. Tumor deposits are defined as foci of carcinoma separated from the main lesion and identified in pericolonic or perirectal fat or the adjacent mesentery. In TNM Ed 7 and 8 tumor deposits are classified as N1c, and they can be diagnosed only with pathological examination of a surgical specimen. Probably the term is used with a different meaning.

Response: Thanks for your professional advice. We have to admit these mistakes. And the wrong expression (tumor deposits) was replaced by residual tumor. 

Reviewer 2 Report

The manuscript is interesting and overall well written.
However there are some critical issues that need to be further stressed:

- the classification of T1 is not differentiated in relation to the infiltration state of the sub-mucosa. I strongly recommend developing the topic by re-evaluating the specimens and inserting this data into the statistical analysis;

- both the endoscopic procedure and surgery have a risk of morbidity and mortality that have not been reported or taken into account in assessing the overall prognosis. The benefits of cancer-related survival may be lost in the complications of major colic resections...

Author Response

The manuscript is interesting and overall well written. However there are some critical issues that need to be further stressed:

- the classification of T1 is not differentiated in relation to the infiltration state of the sub-mucosa. I strongly recommend developing the topic by re-evaluating the specimens and inserting this data into the statistical analysis;

Response: Thanks for your constructive point.  The status of sub-mucosa invasion could potentially influence our outcomes. However, the depth of submucosa invasion (SID) can’t be obtained. Therefore, only the status of submucosa invasion was included in our modified version. As suggested, the corresponding analysis has been added in the manuscript:

Figure 2. Kaplan-Meier cancer-specific survival plots in patients underwent endoscopic removal of invasive neoplasia. (A) All patients; (B) Patients with intramucosal carcinoma (Tis); (C) Patients with submucosal carcinoma (T1). (D) Patients with or without submucosa invasion; (Page 6, line 139)

Table 1. Baseline Characteristics before and after propensity-score matching analysis. (Page 4, line 131)

Characteristics

Before PSM

After PSM

Overall

ET only

ET+Surg

p-value

Overall

ET only

ET+Surg

p-value

N

5805

1214

4591

2428

1214

1214

Sex (%)

Female

2410 (41.5)

485 (40.0)

1925 (41.9)

0.226

989 (40.7)

485 (40.0)

504 (41.5)

0.457

Male

3395 (58.5)

729 (60.0)

2666 (58.1)

1439 (59.3)

729 (60.0)

710 (58.5)

Age (mean (SD))

63.41 (12.05)

64.08 (11.97)

63.23 (12.06)

0.03

64.21 (11.87)

64.08 (11.97)

64.35 (11.76)

0.573

Grade (%)

Well differentiated; Grade I

793 (13.7)

157 (12.9)

636 (13.9)

0.556

315 (13.0)

157 (12.9)

158 (13.0)

0.453

Moderately differentiated; Grade II

1457 (25.1)

290 (23.9)

1167 (25.4)

566 (23.3)

290 (23.9)

276 (22.7)

Poorly differentiated; Grade III

121 (2.1)

25 (2.1)

96 (2.1)

51 (2.1)

25 (2.1)

26 (2.1)

Undifferentiated; anaplastic; Grade IV

72 (1.2)

18 (1.5)

54 (1.2)

27 (1.1)

18 (1.5)

9 (0.7)

Unknown

3362 (57.9)

724 (59.6)

2638 (57.5)

1469 (60.5)

724 (59.6)

745 (61.4)

Histology type (%)

Adenocarcinoma

5654 (97.4)

1176 (96.9)

4478 (97.5)

0.092

2361 (97.2)

1176 (96.9)

1185 (97.6)

0.733

Mucinous adenocarcinoma

23 (0.4)

5 (0.4)

18 (0.4)

7 (0.3)

5 (0.4)

2 (0.2)

Neuroendocrine carcinoma

16 (0.3)

8 (0.7)

8 (0.2)

15 (0.6)

8 (0.7)

7 (0.6)

Signet ring cell carcinoma

4 (0.1)

2 (0.2)

2 (0.0)

4 (0.2)

2 (0.2)

2 (0.2)

Squamous cell carcinoma

4 (0.1)

1 (0.1)

3 (0.1)

1 (0.0)

1 (0.1)

0 (0.0)

Multiply histology types

2 (0.0)

0 (0.0)

2 (0.0)

0 (0.0)

0 (0.0)

0 (0.0)

Unknown

102 (1.8)

22 (1.8)

80 (1.7)

40 (1.6)

22 (1.8)

18 (1.5)

Localization (%)

Right colon

1282 (22.1)

276 (22.7)

1006 (21.9)

0.804

545 (22.4)

276 (22.7)

269 (22.2)

0.678

Left colon

4407 (75.9)

915 (75.4)

3492 (76.1)

1842 (75.9)

915 (75.4)

927 (76.4)

Rectum

116 (2.0)

23 (1.9)

93 (2.0)

41 (1.7)

23 (1.9)

18 (1.5)

T stage (%)

Tis

3496 (60.2)

760 (62.6)

2736 (59.6)

0.061

1497 (61.7)

760 (62.6)

737 (60.7)

0.358

T1

2309 (39.8)

454 (37.4)

1855 (40.4)

931 (38.3)

454 (37.4)

477 (39.3)

T1, Submucosa invasion positive

1062(18.3)

203(16.7)

859(18.7)

402(16.6)

203 (16.7)

199 (16.4)

T1, Submucosa invasion negative

1186(20.4)

227(18.7)

959(20.9)

494(20.3)

227 (18.7)

267 (22.0)

Polyp size (%)

<10mm

1315 (22.7)

260 (21.4)

1055 (23.0)

0.009

519 (21.4)

260 (21.4)

259 (21.3)

0.466

10-19mm

404 (7.0)

65 (5.4)

339 (7.4)

121 (5.0)

65 (5.4)

56 (4.6)

20-29mm

197 (3.4)

40 (3.3)

157 (3.4)

77 (3.2)

40 (3.3)

37 (3.0)

30-39mm

81 (1.4)

19 (1.6)

62 (1.4)

27 (1.1)

19 (1.6)

8 (0.7)

40-49mm

26 (0.4)

4 (0.3)

22 (0.5)

8 (0.3)

4 (0.3)

4 (0.3)

50-mm

31 (0.5)

13 (1.1)

18 (0.4)

27 (1.1)

13 (1.1)

14 (1.2)

Unknown

3751 (64.6)

813 (67.0)

2938 (64.0)

1649 (67.9)

813 (67.0)

836 (68.9)

Variable

ET only group

ET + surgery group

p-value

5-year CSS rate

95%CI

5-year CSS rate

95%CI

Overall

0.958

0.946-0.970

0.974

0.964-0.983

0.017

T stage

Tis stage

0.979

0.968-0.990

0.988

0.981-0.996

0.093

T1 stage

0.921

0.894-0.948

0.950

0.929-0.971

0.047

Submucosa invasion

T1, Submucosa invasion positive

0.947

0.913-0.981

0.950

0.918-0.907

0.552

T1, Submucosa invasion negative

0.924

0.888-0.960

0.949

0.920-0.978

0.949

Polyp size

<10 mm

0.983

0.966-1.000

0.971

0.950-0.992

0.610

10-19 mm

0.976

0.931-1.000

0.960

0.906-1.000

0.770

≥20 mm

0.882

0.808-0.962

0.948

0.893-1.000

0.200

Table 2. 5-year Cancer-specific survival for different treatment modalities among patients with endoscopic removal of invasive neoplasia. (Page 7, line 165)

Table 3. Univariate Cox regression analysis. (Page 8, line 178)

Variable

Hazard ratio

Lower 95% CI

P value

Overall

ET alone

Reference

ET + Surgery

0.61

0.40 - 0.92

0.018

T stage

Tis

ET alone

Reference

ET + Surgery

0.54

0.26 - 1.12

0.099

T1

ET alone

Reference

ET + Surgery

0.61

0.37 - 1.00

0.050

Submucosa invasion

T1, Submucosa invasion positive

ET alone

Reference

ET + Surgery

0.72

0.32 – 1.64

0.435

T1, Submucosa invasion negative

ET alone

Reference

ET + Surgery

0.71

0.36 - 1.40

0.322

Polyp size

<10 mm

ET alone

Reference

ET + Surgery

1.32

0.46 - 3.81

0.607

10-19 mm

ET alone

Reference

ET + Surgery

0.76

0.13 - 4.60

0.766

20- mm

ET alone

Reference

ET + Surgery

0.20

0.02 - 1.76

0.146

Table 4. The migration of lymph nodes and distant tissue/organ among the entire patients with endoscopic polypectomy. (Page 9, line 195)

N stage

M stage

Subgroup

Overall

N0

Probability

N1+N2

Probability

P value

Overall

M0

Probability

M1

Probability

P value

n

5805

5783

99.6210%

22

0.3790%

5805

5768

99.3626%

37

0.6374%

T stage

Tis

3496

3496

100.0000%

0

0.0000%

<0.001

3496

3496

100.0000%

0

0.0000%

<0.001

T1

2309

2287

99.0472%

22

0.9528%

2309

2272

98.3976%

37

1.6024%

T1, Submucosa invasion

Submucosa positive

402

400

99.5025%

2

0.4975%

0.289

402

400

99.5025%

2

0.4975%

0.041

Submucosa negative

494

490

99.1903%

4

0.8097%

494

482

97.5709%

12

2.4291%

Polyp size in mm

<10

1315

1313

99.8479%

2

0.1521%

<0.001

1315

1314

99.9240%

1

0.0760%

<0.001

10-19

404

403

99.7525%

1

0.2475%

404

400

99.0099%

4

0.9901%

20-29

197

195

98.9848%

2

1.0152%

197

194

98.4772%

3

1.5228%

30-39

81

78

96.2963%

3

3.7037%

81

78

96.2963%

3

3.7037%

40-49

26

25

96.1538%

1

3.8462%

26

25

96.1538%

1

3.8462%

>50

31

28

90.3226%

3

9.6774%

31

28

90.3226%

3

9.6774%

Localization

Unknown

3751

3741

99.7334%

10

0.2666%

3751

3729

99.4135%

22

0.5865%

Right colon

1282

1275

99.4540%

7

0.5460%

0.047

1282

1267

98.8300%

15

1.1700%

<0.001

Left colon

4407

4394

99.7050%

13

0.2950%

4407

4388

99.5689%

19

0.4311%

Rectum

116

114

98.2759%

2

1.7241%

116

113

97.4138%

3

2.5862%

CEA

Negative

424

420

99.0566%

4

0.9434%

<0.001

424

421

99.2925%

3

0.7075%

<0.001

Positive

120

112

93.3333%

8

6.6667%

120

95

79.1667%

25

20.8333%

Unknown

5261

5251

99.8099%

10

0.1901%

5261

5252

99.8289%

9

0.1711%

- both the endoscopic procedure and surgery have a risk of morbidity and mortality that have not been reported or taken into account in assessing the overall prognosis. The benefits of cancer-related survival may be lost in the complications of major colic resections...

Repsonse: Thanks for your constructive advices. In order to evaluate other complication reasons that could contribute to the survival analysis. The overall survival analysis were performed, as shown in supplementary Figure S 1 and S2.

Supplementary Figure S1. Kaplan-Meier overall survival plots in patients underwent endoscopic removal of invasive neoplasia. (A) All patients; (B) Patients with intramucosal carcinoma (Tis); (C) Patients with submucosal carcinoma (T1). (D) Patients with or without submucosa invasion;

Supplementary Figure S2. Kaplan-Meier overall survival plots in patients underwent endoscopic removal of invasive neoplasia. (A) Invasive polyp size less than 10 mm; (B) Invasive polyp size between 10 mm and 20 mm; (C) Invasive polyp size larger than 20 mm.

Round 2

Reviewer 2 Report

The Authors adequately satisfied the review for submucosal involvement.

Possible complications after surgical or endoscopic resection should instead be discussed in the main text and not only in the supplementary materials.

Author Response

Dear Reviewer, thanks for this remark. We inserted the follwoing paragraph into the discussion part: "In summary possible complications after endoscopic resection are the misidentification of the T stage and polyp size, and periprocedural complications as bleeding and perforation. Complications after surgical resection of Tis and T1 malignancies are the over-extended operation with subsequent functional impairments and possible long-time complications of the hospital stay and the operation mode like surgical side infections or incisional hernia."